# Lacking Pace but Not Precision: Age-Related Information Processing Changes in Response to a Dynamic Attentional Control Task

**DOI:** 10.3390/brainsci10060390

**Published:** 2020-06-19

**Authors:** Anna Torrens-Burton, Claire J. Hanley, Rodger Wood, Nasreen Basoudan, Jade Eloise Norris, Emma Richards, Andrea Tales

**Affiliations:** 1Division of Population Medicine, School of Medicine, Cardiff University, Cardiff CF14 4XN, UK; Torrens-BurtonA@cardiff.ac.uk; 2Department of Psychology, Swansea University, Swansea SA2 8PP, UK; c.j.Hanley@swansea.ac.uk (C.J.H.); r.l.wood@swansea.ac.uk (R.W.); nasreen.s.basoudan@gmail.com (N.B.); 3Department of Psychology, University of Bath, Bath BA2 7AY, UK; jn528@bath.ac.uk; 4Centre for Innovative Ageing, Swansea University, Swansea SA2 8PP, UK; e.v.richards@swansea.ac.uk

**Keywords:** reaction time, intra-individual variability, subjective memory, healthy ageing, cognitive impairment

## Abstract

Age-related decline in information processing can have a substantial impact on activities such as driving. However, the assessment of these changes is often carried out using cognitive tasks that do not adequately represent the dynamic process of updating environmental stimuli. Equally, traditional tests are often static in their approach to task complexity, and do not assess difficulty within the bounds of an individual’s capability. To address these limitations, we used a more ecologically valid measure, the Swansea Test of Attentional Control (STAC), in which a threshold for information processing speed is established at a given level of accuracy. We aimed to delineate how older, compared to younger, adults varied in their performance of the task, while also assessing relationships between the task outcome and gender, general cognition (MoCA), perceived memory function (MFQ), cognitive reserve (NART), and aspects of mood (PHQ-9, GAD-7). The results indicate that older adults were significantly slower than younger adults but no less precise, irrespective of gender. Age was negatively correlated with the speed of task performance. Our measure of general cognition was positively correlated with the task speed threshold but not with age per se. Perceived memory function, cognitive reserve, and mood were not related to task performance. The findings indicate that while attentional control is less efficient in older adulthood, age alone is not a defining factor in relation to accuracy. In a real-life context, general cognitive function, in conjunction with dynamic measures such as STAC, may represent a far more effective strategy for assessing the complex executive functions underlying driving ability.

## 1. Introduction

Executive function-related reaction time (RT) and its individual variability (IIV) are regarded as behavioural markers of central nervous system integrity [1,2,3,4]. In relation to ageing, RT slowing and increased IIV are commonly described behavioural characteristics of normal, healthy processes [1,5], with disproportionate levels of decline associated with functional deficits and disease, such as mild cognitive impairment (MCI), vascular cognitive impairment (VCI), and Alzheimer’s disease (AD) [6,7,8,9,10,11,12,13,14,15], as well as conditions such as mild traumatic brain injury (mTBI) [16]. It is common, therefore, for RT tests to be included in clinical diagnostic batteries.

Generally, in such tests, RT is measured over multiple sequential trials in which a response to the same given stimulus is required. Blocks of discrete trials are delivered, often with uniform task difficulty, and are analysed by focusing on each separate event or averaging responses over time [3,11,12,15]. However, the intricacies of visual perception in the real-world are far more complex than that represented by the measurement of RT in this way. For example, mediated by attentional control processes [17], representations are constantly updated to inform the individual of the crucial elements in scenes of varying complexity [18,19], in driving for example [16,20]. This means that approaches that adopt discrete trials cannot mimic the manner in which attentional control operates, nor can they represent RT in its real-life context, thus limiting the validity of the inferences that can be made when utilizing such methods.

### 1.1. Attentional Control

Attentional control processes are associated with a network involving prefrontal and parietal regions, which govern target detection and response inhibition, under conditions of varying difficulty and demand for resources [21,22], all of which in real-life would contribute to a measure of RT at any given point. Although there is some evidence to suggest that for lower task demands older adults can maintain relatively intact performance (as a result of an ability to increase recruitment of available neural resources to stabilise performance), age-related decreases in activity tend to occur with higher task demands [23,24,25] because available processing resources are saturated [17]. Consequently, age-related failure to modulate activity in response to changes in task demand may indicate an inability to dynamically allocate attentional control in response to change. Furthermore, although difficulty can be varied in common attention-related RT tasks such as those examining visual search, inhibition of return, alerting, and orienting [14,26,27,28,29,30], it is not typically investigated with respect to individual capability (e.g., accommodating differences in what individual participants find challenging), which is an important consideration when devising tasks for use with a highly heterogeneous population, such as older adults.

Typical RT studies also simply examine the relationship between error production and RT, in order to determine whether a speed/accuracy trade-off is influencing the results [26,28,30,31]. In reality, however, the interplay between speed and accuracy is likely to be far more relevant to real life environmental processing. When responding appropriately to environmental change, a certain level of accuracy is required as well as speed. It is likely that the ability to maintain accuracy, whilst also maintaining speed, is a more adaptive behaviour than the ability to respond quickly to a given stimulus that is repeatedly presented, with no change in resource requirements. Examining RT in isolation from the integrity of attentional control, and thus the above-mentioned functions, may lead to the under- or over- estimation of an individual’s functional ability; a factor which may be of particular relevance to the assessment of driving ability [32] as this behaviour is highly dependent upon the integrity of attentional control components such as selective attention, attentional switching, and the inhibition of irrelevant visual information [33,34,35,36,37]. As driving cessation is widely considered to be a major life transition which can have a significant impact on the health and well-being of older drivers [38,39,40], it is vital that such ability is assessed appropriately [32,38,41].

### 1.2. Study Rationale

In this study, we examined ageing-related attentional control (i.e., how much information can be processed over a specific time period, at a given level of accuracy) using a novel task, designed to address the outlined issues with traditional tests; namely the computer-based Swansea Test of Attentional Control (STAC). The STAC (developed by Carter and Wood, and outlined in Hanley and Tales, (2019) [42] comprises selective attention, task monitoring, and response inhibition components of attentional control based on the supervisory attentional system model [22], making it ideal to simulate the complex demands of continuous environmental monitoring and interaction, within a single test. A flexible algorithm designed to track performance (Parameter Estimation by Sequential Testing, PEST; [43]), calibrates the speed at which a given degree of response accuracy can be maintained. STAC also relies on the continuous presentation of stimuli, as opposed to delivering discrete trials in blocks that systematically vary task demands [17,44]. Therefore, compared to other attentional control tasks in isolation, STAC has the benefit of being more holistic in relation to the demands of stimulus engagement in everyday life. Furthermore, the stability and validity of the STAC task have previously been assessed, with good test–retest reliability and strong correspondence between STAC final speed and RTs from a standard Flanker task (for further insight and information about this test, see Hanley and Tales, (2019) [42]). Unlike traditional tests, which focus on single elements of attentional control in isolation, STAC is more holistic and integrates components of selective attention, task monitoring, and response inhibition. Furthermore, while there is overlap in the constructs tested, standard tasks such as the Flanker test (known for its ability to assess response inhibition) require the researcher to set parameters (e.g., stimulus presentation speed) in advance, and often lack flexibility because such values are fixed, meaning the participant is forced to struggle to respond or is not sufficiently challenged. Use of the PEST algorithm as part of STAC, which calibrates speed on the basis of prior responses, has the distinct advantage of ensuring that the task is performed in accordance with an individual’s capabilities. Therefore, the task remains difficult and will challenge the participant as speed is re-adjusted during subsequent PEST cycles (either upwards or downwards, to define their threshold, in the event that the present speed can be exceeded or is too challenging and cannot be sustained).

The comparison of STAC to the Flanker task corresponds to our previous pilot work. While the Flanker task output equates to reaction time (RT), the STAC final speed threshold reflects a participants’ ability to perform well under conditions of higher stimulus presentation speeds. Nonetheless, both measures represent efficiency of processing with regard to speed. Where RT from incongruent Flanker trials (representing conditions of maximum difficulty) is assessed in relation to STAC speed, there is strong correspondence between these measures (*r* (24) = −0.650, *p* = 0.001). The correlation indicates that as STAC speed increases, Flanker RT decreases, thus signifying performance improvement between the measures is aligned. Therefore, STAC is regarded as comparable to Flanker in relation to outcome measures but has distinct advantages compared to such standard tasks, as outlined above.

Adopting this novel task, the primary aim of the study was to examine attentional control in older compared to younger adults. Within the scope of test complexity and task difficulty, whilst task difficulty is not a factor under direct manipulation, with discrete levels of complexity, it corresponds to the speed of performance (spm); whereby performing the task at a higher speed would increase the demand for processing resources. Accordingly, higher speed thresholds represent the ability of participants to successfully complete the task under conditions of increased difficulty.

We also aimed to determine whether attentional control varied with respect to general cognitive function (using the Montreal Cognitive Assessment (MoCA)) [45], typically used to define an older adult control group. For the older adults, we also examined performance with respect to subjective feelings of memory function (using the Memory Function Questionnaire (MFQ) [46], cognitive reserve (using the National Adult Reading Test, NART score as a proxy) and educational level [47,48,49,50,51,52]. Finally, a mix of male and female older adult participants was included, all with depression and anxiety within normal levels.

In relation to these aims, we predicted that attentional control would be significantly poorer in older compared to younger adults, as determined by speed of performance on the STAC task; and that high levels of cognitive control would be significantly associated with high cognitive function, high cognitive reserve, and lower levels of perceived abnormal memory. We also investigated whether attentional control varied with respect to gender and sub-clinical levels of depression and anxiety [13,53,54].

By obtaining this range of measurements we were well positioned to determine how, if at all, attentional control is related to the level of objectively and subjectively measured cognitive function, within individuals. This is a crucial consideration as older adults with ostensibly normal cognition for their age and educational level, and some individuals with subjective cognitive decline (SCD) [55], can show considerable variation in RT and attention. Some individuals may even present with levels of detrimental functional change more typical of MCI, vascular cognitive impairment (VCI), and Alzheimer’s disease (AD) [11,12,13,14].

## 2. Materials and Methods

### 2.1. Participants

Community-dwelling older adults (*n* = 90; age 50–79 years; 38 males, 52 females) were recruited through advertisements at older adult social clubs, via local newspapers, word of mouth, and via the older adult research volunteer database (Department of Psychology, Swansea University), with no specified upper age limit. Younger adults, aged between 18 and 30 years and thus typical of younger control groups in ageing studies, (*n* = 82; age 18–27 years; 32 males, 50 females) were recruited via poster, social networking, and email advertisements throughout Swansea University. Testing was conducted within dedicated research rooms within the Department of Psychology at Swansea University. Ethical approval was granted by the Swansea University Department of Psychology Research Ethics Committee (Reference number: 01-20/2-2015-1), and the study was conducted in accordance with the principles of the Declaration of Helsinki. Written, informed consent was given by each participant, and all participants were debriefed after participation.

All participants self-reported to be in good general health, with no history of serious head injury, mental health problems, or neurological impairment. None of the participants had been to see their general practitioner (GP) or memory services about change in their cognitive function or concern about it, and all participants had normal or corrected-to-normal vision. The groups were matched with respect to gender distribution and educational level (Table 1). None of the participants had significant levels of depression or anxiety, as indicated by a score of 9 or below on the Patient Health Questionnaire (PHQ-9) [56] and a score of 5 or below on the Generalized Anxiety Disorder 7-item (GAD-7) [57] (using pen and paper-based versions of both of these tests) and individuals with a history of significant depression or anxiety were excluded from this study. Although medication use could not be controlled, individuals taking drugs likely to affect cognition (such as benzodiazepines) were excluded from recruitment.

### 2.2. Materials and Procedures

Cognitive function was assessed using the pen and paper-based Montreal Cognitive Assessment (MoCA) [45]. For the older adults, subjective memory function was assessed by the pen and paper-based Memory Functioning Questionnaire (MFQ) [46] (in which lower scores represent a greater degree of perceived memory impairment). The NART was administered to the older adult participants as a proxy for cognitive reserve [47,48,49]. The order of these tests was not counterbalanced, and all tests were administered and scored according to current guidelines. The computer-based STAC test was then administered.

Symbols are 36.2 mm across, subtending 3.6 degrees of visual angle on a 19-inch monitor at 52 cm viewing distance. A target is identified within the 3 × 3 matrix of symbols (right). When a matching symbol appears amongst the three columns of the search array (left), which scroll up the left-hand side of the screen, participants press the spacebar as the symbol crosses behind the red line. The task is to identify the target within a 3 × 3 matrix of symbols on the right and search for matching symbols amongst an array of three columns of symbols (Figure 1). The target changes throughout the task such that participants must remain vigilant in order to consistently update their search criteria, while simultaneously monitoring the search array to identify matching items and inhibit irrelevant symbols. The target changes every 19 s but is delayed if the current target appears in the search array (e.g., the target will not change to a different symbol if the current target is on screen). In such instances, the corresponding time lapse is added to the total run time (initially set to 180 s, resulting in ~9 target changes for all participants). Speed (measured in symbols per minute per column; abbreviated to “spm”) is adjusted to maintain accuracy around a commonly adopted 75% correct criterion [58], using the PEST algorithm. The task begins at a speed of 60 spm. After a minimum of 4 target changes, speed is calibrated on the basis of performance accuracy, increasing or decreasing with a step-size between 20 and 60 spm. The participants’ thresholds are the average speed at which the task is performed across the duration of the task.

### 2.3. Data Analysis and Results

With respect to demographics, Mann–Whitney analysis revealed no significant difference in MoCA score or mean years in education between young and older adult groups (*p* > 0.05). Depression level (U = 1911.5, *p* < 0.001, *r* = 0.37) and anxiety level (U = 1638, *p* < 0.001, *r* = 0.32) were significantly greater for young adults compared to older adults.

Non-parametric analyses were performed as the STAC data were not normally distributed.

Mann–Whitney U analysis revealed no significant difference in mean accuracy between the older and younger adults (U = 3580.5, *p* = 0.74) and for both the younger and older groups this did not vary significantly with respect to gender ((male U = 785.5, *p* = 0.89) and (female U = 946.5, *p* = 0.73)). There was, however, a significant difference in spm, with younger adults able to process significantly more symbols per minute per column than the older adults, performing the task significantly faster (U = 2059.5, *p* < 0.001; Cohen’s effect size = 0.884) (Table 2 and Figure 2). There was no significant difference in spm between male and female participants, for both the younger (U = 602.5, *p* = 0.06) and older groups (U = 978.5, *p* = 0.94), respectively.

For the older adults, Spearman’s rho analyses revealed a significant correlation between spm and MoCA score (*r* = 0.25, *p* = 0.018), with better performance associated with higher MoCA scores (higher levels of general cognitive function). In contrast, there was no significant correlation between spm and MoCA score (*p* > 0.05) for the younger adults.

There was also a significant negative correlation between spm and age (*r* = −0.29, *p* = 0.005) (as age increased, spm decreased) for the older adults (Figure 3). However, there was no significant correlation between age and MoCA score (*p* > 0.05). For the younger adults, there was no significant correlation between spm and age (*p* > 0.05), and no significant correlation between age and MoCA score (*p* > 0.05). For the older adults, spm was not significantly correlated with MFQ or NART, and there was no significant correlation between MFQ and MoCA scores (all *p*-values > 0.05). For both the younger and older adults, spm was not significantly correlated with anxiety, depression, or educational level (all *p*-values > 0.05).

## 3. Discussion

In this study, we used the STAC test to examine the complex interplay between the speed and accuracy of information processing in older compared to younger adults. As predicted, the requirement to maintain a stimulus accuracy response of at least 75% (actually achieving approximately 84% in this study) resulted in a reduction in the amount of processing possible within a given time period. Specifically, this was characterised by a significant decrease in the speed at which the symbols per minute per column could be processed, for the older compared to the younger group. The ability to modulate activity to varying task resource demands and inhibit interference while maintaining a high level of processing speed for a given accuracy of performance (attentional control), therefore, appears significantly less efficient in older adulthood. This is likely a result of a reduced ability to dynamically allocate attention, such that supply is equal to demand. These results, which were independent of gender, are in line with well-established evidence showing that higher demand for resources, under conditions of greater complexity, causes issues in relation to the efficacy of information processing with age [17,23,24,25]. Unlike previous studies, we employed methodology that allowed for within-group variation in respect to individual capability (i.e., accommodating differences in what participants find challenging), thus removing a potential confounding factor present in previous studies with variation in task difficulty; an important consideration when devising tasks for use with a highly heterogeneous older adult population.

We were also interested in examining any relationship between general cognition measured using the MoCA [45] in individuals who had not reported any cognitive changes or concerns to their general practitioner/memory services, and STAC performance. For the older adults there was a significant correlation between spm and MoCA score, with higher levels of general cognition function associated with better performance; a significant negative correlation between spm and age with spm decreasing as age increased, but there was no significant correlation between age and MoCA score. This pattern of results suggests that the reduction in speed of performance as age increases may be compensated for (at least within the age range of the participants in this study), by higher levels of objectively measured levels of cognitive function. The lack of a significant correlation between spm and MFQ score indicates that attentional control is not however associated with, or compensated for, by perceived level of cognitive function.

Attentional control in older adults was also not a function of cognitive reserve, (using the proxy measure of NART score), although one could argue that NART score was not representative of the complexity of cognitive reserve, nor of all its components. In contrast to the older adults, there was no significant correlation between spm and age, or spm and MoCA score, for the younger adults. For both the younger and older adults there was no significant correlation between spm and anxiety, depression or educational level, indicating that attentional control was not associated with such factors at the levels or ranges encountered in the present study.

With respect to older adults, the significant correlation between spm and MoCA score, significant negative correlation between spm and age, and the lack of significant correlation between age and MoCA score, indicates that chronological age alone is unlikely to be useful in determining real life behaviour such as driving. This supports the idea that functional ageing, as opposed to chronological ageing, may be a more appropriate indicator of real-life ability. The results indicate that although one gets slower with increasing age, accuracy is still maintained. This is likely true up to a point, or certain age; after which it may no longer be appropriate for older adults to engage in driving behaviour that is more “resource intensive” (such as driving on motorways or at night, which requires higher speeds and increased vigilance). Although the correlation between MoCA score and STAC performance (spm) may provide some insight into how well an individual can manage these demands, it may be more appropriate if lower scores were used to prompt more in-depth cognitive assessment, such as use of STAC, to assess driving behaviour [16,20,32,38,41], particularly as driving cessation is widely considered to be a major life transition, and can have a significant impact on the health and well-being of older drivers [39,40].

### 3.1. Potential Study Limitations

We have previously shown that older adults (mean age 66.5 years) are able to comfortably perform the STAC at a length of 300 s [42]. However, due to the novel nature of the task, the “optimal” duration for both younger and older adults is yet to be determined. In this instance, the task was only run for a short amount of time (180 s), which may have favoured performance in the younger group more so than the older group, because for the latter it has been found to improve with continued exposure. Therefore, the distinction between groups may be diminished at longer intervals. Furthermore, although the PEST algorithm should naturally converge on a participant’s optimal speed, given sufficient time, it would certainly be an intriguing prospect to assess performance capability where the start speed of the experiment was founded on the basis of an individual’s threshold (established during practice, for example).

### 3.2. Future Studies

Without accompanying neuroimaging data, the study is not able to make direct inferences relating to the efficiency of neural processes supporting attentional control. In future, it would be of interest to use task-based fMRI to assess the availability of processing resources in given regions or networks (prefrontal/parietal). Use of diffusion-weighted imaging to assess the integrity of key tracts related to attention (genu/forceps minor, superior longitudinal fasciculus) could also be incorporated. Additional electroencephalographic (EEG) analysis both in the time-domain and in the time-frequency domain would also reveal the recruitment of additional cognitive resources required for the maintenance of the accuracy with increasing age. The examination of the variations in task difficulty and complexity based on the individual threshold of a single participant (e.g., a variation of a specific % of spm), would also provide a unique insight into the neural underpinnings of this novel task, age-related distinctions, and individual differences.

The STAC represents a complex task that requires good attention control to optimise efficient information processing. These processes rely on sub-cortical white matter integrity for speed and efficiency of processing, and the pre-frontal cortex to “supervise” the allocation of attentional resources. These cerebral regions are vulnerable to the decelerative forces present in traumatic brain injury and often underpin weaknesses of cognition reported by patients, even after relative minor head trauma. However, identifying attentional failures during a clinical assessment can be difficult [58]. This probably accounts for the dislocation, often seen when a patient performs well on clinical tests of cognition yet complains of intrusive lapses of working memory and other cognitive failures in everyday life. This disparity is assumed to reflect the different attentional loadings present in a clinical assessment versus everyday life. As the STAC replicates the changing attentional demands encountered in the cognitive control of everyday activities, it may also be an important tool for the investigation of head injury with respect to the determination functional integrity and response to intervention.

Furthermore, cognitive function was only assessed using the MoCA [45] and thus only a composite score of performance over several cognitive domains, namely short-term memory, visuospatial ability, executive function, attention, concentration and working memory, language, and orientation to time and place. Future studies will involve the examination of a wider range of specific cognitive functions in order to examine the links between cognitive function and attentional control, information processing speed, and accuracy.

Finally, it would be of value to administer the task for a range of durations to assess age-related differences under conditions of varying demands on sustained attention. Furthermore, the age-range was relatively large in the present study, and although there was no specified upper age limit our older adult group contained no-one over the age of 79 years and future studies should focus on a smaller age range or sub-divide the older age group into different cohorts (e.g., 50–65, 66–80) and to include individuals above the age of 80 in order to assess age effects in more detail.

## Figures and Tables

**Figure 1 brainsci-10-00390-f001:**
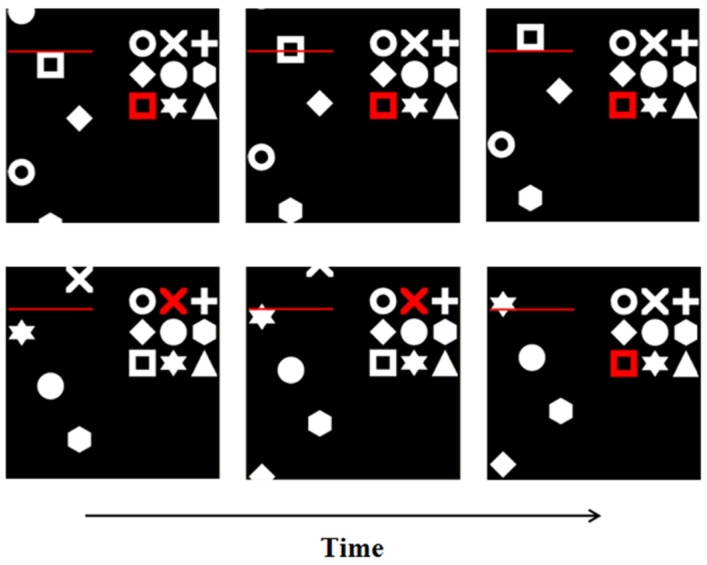
Swansea Test of Attentional Control. Participants view a 3 × 3 matrix of symbols (screen right) and a continuous stream of symbols (screen left). Top: The symbols on the left scroll up the screen. Where an oncoming symbol matches the target (highlighted in red, on the right), participants respond when that shape crosses the red line (as opposed to before or after). Bottom: The target changes every 19 s, such that participants must remain vigilant and consistently attend to both elements of the task to ensure successful performance.

**Figure 2 brainsci-10-00390-f002:**
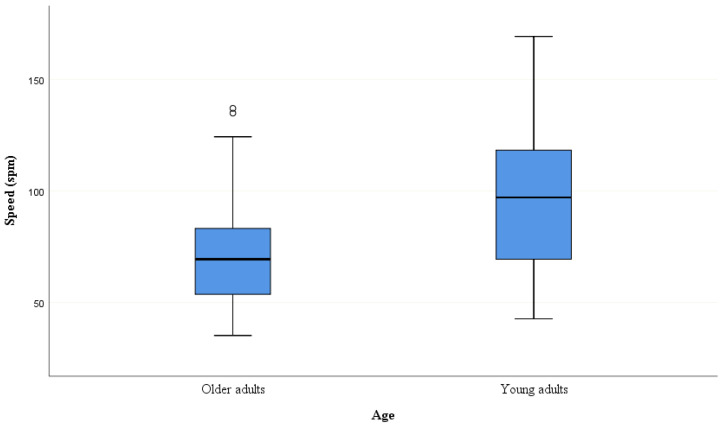
Speed (symbols per minute per column; spm) between young and older adults.

**Figure 3 brainsci-10-00390-f003:**
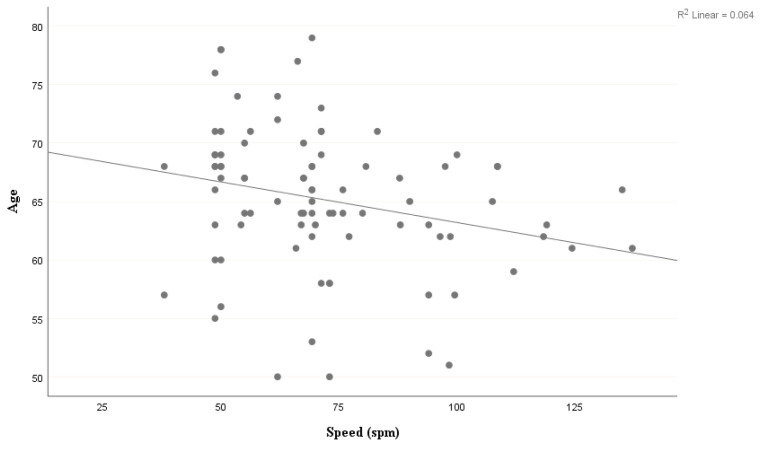
Correlation between speed (spm) and MoCA score in older adults.

**Table 1 brainsci-10-00390-t001:** Baseline group mean demographics for the younger and older adult groups. Standard deviation in parenthesis.

	Age (Years)	Education Level (Years)	NART	MoCA Score	MFQ (Total Score)	PHQ-9	GAD-7
Young Adults	20 (2.1)	14.7 (3.5)	-	27.0 (2.1)	-	6.3 (4.4)	4.8 (4.3)
Older Adults	65 (6.1) Range 50–79	15.1 (4.8)	41.4 (5.4)	27.4 (2.2)	292.8 (48.8)	3.2 (3.1)	2.3 (2.5)

All data were analysed using SPSS version 22 (IBM Corp., Armonk, NY, USA).

**Table 2 brainsci-10-00390-t002:** Symbols per minute per column (spm) and level of accuracy (%) for younger and older adults. Standard deviation in parentheses.

	Mean spm	% Accuracy
Young Adults	94.78 (29.69)	83.82 (9.15)
Older Adults	71.74 (22.26)	83.16 (13.63)

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
