# Peer review of "Lacking Pace but Not Precision: Age-Related Information Processing Changes in Response to a Dynamic Attentional Control Task"

_brainsci, 2020, doi:10.3390/brainsci10060390_

Round 1
Reviewer 1 Report
The Authors presented a study in which the Swansea Test of Attentional Control (STAC) is used to assess differences in attentional control between two groups of participants that differed in age (younger and older adults). The results showed that the older (than younger) adults were slower in performing the task (i.e., symbols per minutes, spm), and that such a measure was positively correlated with their general cognitive functioning, but negatively correlated with their age. The manuscript is well-organized, the language is appropriate, and the proposal of a new task to assess attentional functions is quite interesting. That said, I encourage the Authors to take into consideration a series of points that could strengthen the work if integrated with their current proposal.
Major points:
- Abstract. The opening sentence of the abstract refers to changes in the visual system as a function of aging and the possible impact on a complex ability such as driving. The former point is not discussed in the main text. I suggest the Authors to slightly change their opening line or incorporate it in the introduction of the manuscript. Similarly, the second point is briefly mentioned at the end of the introduction, but more importantly, in the discussion, it is not sufficiently linked to the previous paragraph of that section.
- Introduction. The Authors referred several times to “tests” (e.g., page 2, line 48, 68, 88) used to assess executive and attentional functions. I would suggest reporting a few examples of such tests/tasks indicating the cognitive processes or domains they aimed to investigate (e.g., what subcomponents of attentional controls).
- As indicated in my previous comment, the reference to “driving ability” (page 2, line 81) would require contextualization and integration.
- I appreciate the dedicated section to the study rationale reporting the aims of the study. That said, a definition of the subcomponents of attentional control (and maybe a reference to a model of attention that Authors are aiming to assess) would strengthen this section (page 3, line 90).
- The sentence of the Authors “Within the scope of test complexity and task difficulty” (page 3, line 102) is not clear to me since there is no manipulation of task difficulty or complexity in their study (there is no such a factor considered in the analysis and result section). In this regard, it would also be interesting (in future studies) to consider a variation of difficulty/complexity based on the individual threshold of a single participant (e.g., a variation of a specific % of spm).
- Materials and Procedures. What is the rationale behind the age of the two groups of participants?
- The reader can easily understand that the STAC is a computer-based task, but this information was not specified in the text. Since several neuropsychological tests and questionnaires for cognitive and attentional functions are pencil and paper-based, this information should be indicated. Do all the tests (e.g., MoCA, PHQ-9, etc.) used were computer-based? Where was the testing conducted (e.g., university, in different places based on the participants’ group)? How far were the participants from the monitor? What was the size of the stimuli on the monitor?
- What was the initial speed of the scrolling stimuli presented on the left side of the monitor? What was the criterion for the changes in the speed of scrolling? How many trials (e.g., target changes) were completed for each participant? Did the number of trials differ between groups? Was this measure taken into consideration for the analysis, besides the accuracy and spm?
- Discussion. Since the only significant correlation with other cognitive tests was found for the MoCA, I would suggest the Authors deepen the cognitive constructs assesses by this test and discuss it in relation to their task.
- The Authors briefly mentioned that the performance in the STAC had been previously compared with the performance in the Flaker task (page 3, line 99). Given that no additional tasks were included to assess the attentional functions of the participants in the present study, I would suggest discussing such findings instead of inviting the reader to look at that paper for further insight. What is the relationship between the attentional constructs investigated by the STAC and those assessed with typical tasks/tests available in literature?
- Future studies. The Authors indicated in this section that they intend to implement the task during fMRI scanning. I would suggest the Authors explore the vast electrophysiological (EEG) literature on attentional control. Given the very high temporal resolution of EEG, this technique would be ideal for assessing differences concerning the temporal aspects of the performance. Analysis both in time-domain (using ERPs: P300, CNV component) and in the time-frequency domain would point out the recruitment of additional cognitive resources required for the maintenance of the accuracy with increasing age.
Minor points:
- An image that illustrates the timing of the experiment (e.g., one trial) would be more useful instead of a single static image. E.g., a series of frames that illustrate the scrolling up of the left side of the screen, or the target change on the right side of the screen (along with the approximate/mean timing: target change every 19s).
- A visualization of the results would help the reader in understanding the results. For example, barplots or boxplots could be used for non-parametric analyses, while scatterplots could be used for visualizing the sets of correlation proposed.
Author Response
Thank you for reviewing our paper and for your very helpful comments.
The italics in our response represent the changes made within the manuscript. All changes within the manuscript are illustrated using ‘track-changes’
Major points:
- Abstract. The opening sentence of the abstract refers to changes in the visual system as a function of aging and the possible impact on a complex ability such as driving. The former point is not discussed in the main text. I suggest the Authors to slightly change their opening line or incorporate it in the introduction of the manuscript. Similarly, the second point is briefly mentioned at the end of the introduction, but more importantly, in the discussion, it is not sufficiently linked to the previous paragraph of that section.
Thank you for pointing this out. We have adjusted the abstract as suggested.
Abstract: Age-related decline in information processing can have a substantial impact on activities such as driving.
Similarly, we have better linked our discussion about driving with the results described in the preceding section of the discussion.
With respect to older adults, the significant correlation between spm and MoCA score, significant negative correlation between spm and age, and the lack of significant correlation between age and MoCA score, indicates that chronological age alone is unlikely to be useful in determining real life behavior such as driving.
We have also better linked this information within the Introduction and added further relevant references.
(line 82) Examining RT in isolation from the integrity of attentional control, and thus the above-mentioned functions, may lead to the under- or over- estimation of an individual’s functional ability; a factor which may be of particular relevance to the assessment of driving ability [26] as it is highly dependent upon the integrity of attentional control components such as selective attention, attentional switching, the inhibition of irrelevant visual information [Callaghan et al 2017, Stinchcombe et al 2011, Tsang 2013; Roca et al 2011; McManus et al 2017;]. As driving cessation is widely considered to be a major life transition which can have a significant impact on the health and well-being of older drivers [49, 50], it is vital that it is assessed appropriately [26, 47, 48].
- Introduction. The Authors referred several times to “tests” (e.g., page 2, line 48, 68, 88) used to assess executive and attentional functions. I would suggest reporting a few examples of such tests/tasks indicating the cognitive processes or domains they aimed to investigate (e.g., what subcomponents of attentional controls).
This has been addressed, e.g. please see line 69:
Furthermore, although difficulty can be varied in common attention-related RT tasks such as those examining visual search, inhibition of return, alerting and orienting [Tales et al 2004; Tales et al 2011; Bayer et al 2014; Eimer 2014; Torrens-Burton et al 2017; Vaportzis et al 2103], it is not typically investigated with respect to individual capability (e.g. accommodating differences in what...
As indicated in my previous comment, the reference to “driving ability” (page 2, line 81) would require contextualization and integration.
This has been addressed, please see line 82.
Examining RT in isolation from the integrity of attentional control, and thus the above-mentioned functions, may lead to the under- or over- estimation of an individual’s functional ability; a factor which may be of particular relevance to the assessment of driving ability [26] as it is highly dependent upon the integrity of attentional control components such as selective attention, attentional switching, the inhibition of irrelevant visual information [Callaghan et al 2017, Stinchcombe et al 2011, Tsang 2013; Roca et al 2011; McManus et al 2017;]. As driving cessation is widely considered to be a major life transition which can have a significant impact on the health and well-being of older drivers [49, 50], it is vital that it is assessed appropriately [26, 47, 48].
- I appreciate the dedicated section to the study rationale reporting the aims of the study. That said, a definition of the subcomponents of attentional control (and maybe a reference to a model of attention that Authors are aiming to assess) would strengthen this section (page 3, line 90).
This has been addressed please see line 107...
...comprises selective attention, task monitoring, and response inhibition components of attentional control based on supervisory attentional system model [Cieslik et al 2014], making it ideal to simulate the complex demands of continuous environmental monitoring and interaction, within a single test.
- The sentence of the Authors “Within the scope of test complexity and task difficulty” (page 3, line 102) is not clear to me since there is no manipulation of task difficulty or complexity in their study (there is no such a factor considered in the analysis and result section). In this regard, it would also be interesting (in future studies) to consider a variation of difficulty/complexity based on the individual threshold of a single participant (e.g., a variation of a specific % of spm).
While task difficulty is not a factor under direct manipulation, with discrete levels of complexity, it corresponds to the speed of performance (spm); whereby performing the task at a higher speed would increase the demand for processing resources. Accordingly, higher speed thresholds represent the ability of participants to successfully complete the task under conditions of increased difficulty.
The PEST algorithm should naturally converge on a participant’s optimal speed, given sufficient time. However, it would certainly be an intriguing prospect to assess performance capability where the start speed of the experiment was founded on the basis of an individual’s threshold (established during practice, for example).
Materials and Procedures. What is the rationale behind the age of the two groups of participants?
This has been addressed:
Community-dwelling older adults (n = 90; age 50-79 years; 38 males, 52 females) were recruited through advertisements at older adult social clubs, via local newspapers, word of mouth, and via the older adult research volunteer database (Department of Psychology, Swansea University), with no specified upper age limit. Younger adults, aged between 18 and 30 years and thus typical of younger control groups in ageing studies (n= 82; age 18-27 years; 32 males, 50 females) were recruited via poster, social networking, and email advertisements throughout Swansea University.
The reader can easily understand that the STAC is a computer-based task, but this information was not specified in the text. This has been addressed: e.g. The computer-based STAC test was then administered, now appears within the text.
- Since several neuropsychological tests and questionnaires for cognitive and attentional functions are pencil and paper-based, this information should be indicated.
Yes, thank you for pointing this omission out. This has been addressed within the abstract and the main body (please see below).
- Do all the tests (e.g., MoCA, PHQ-9, etc.) used were computer-based?
No, they were pen and paper-based. This has now been highlighted within the text see line 204 and a score of 5 or below on the Generalized Anxiety Disorder 7-item (GAD-7) [45] (using pen and paper-based versions of both of these tests)
- Where was the testing conducted (e.g., university, in different places based on the participants’ group)?
Thank you for pointing out this omission. This has now been corrected within the text see line 187: Testing was conducted within dedicated research rooms within the Department of Psychology at Swansea University.
- How far were the participants from the monitor? What was the size of the stimuli on the monitor?
Thank you for pointing out this omission. This has been addressed. See line 226: Symbols are 36.2 mm across, subtending 3.6 degrees of visual angle on a 19-inch monitor at 52cm viewing distance.
- What was the initial speed of the scrolling stimuli presented on the left side of the monitor? What was the criterion for the changes in the speed of scrolling? How many trials (e.g., target changes) were completed for each participant? Did the number of trials differ between groups? Was this measure taken into consideration for the analysis, besides the accuracy and spm? This has now been addressed and the following appears within the manuscript.
The initial speed was set to 60 spm (line 166). Speed was determined in line with accuracy, with participants needing to fulfil the 75% correct criterion in order for the task to increase in speed. Where this was not achieved, the task speed slowed down (line 164-5). Given the continuous nature of data acquisition (line 94-5), the STAC task does not operate on a set number of trials but the length of the task was set to 180 seconds (line 163), resulting in ~9 target changes for all participants.
Discussion. Since the only significant correlation with other cognitive tests was found for the MoCA, I would suggest the Authors deepen the cognitive constructs assesses by this test and discuss it in relation to their task.
This has been addressed: please see line 437:
Furthermore, cognitive function was only assessed using the MoCA [30] and thus only a composite score of performance over several cognitive domains, namely short-term memory, visuospatial ability, executive function, attention, concentration and working memory, language and orientation to time and place. Future studies will involve the examination of a wider range of specific cognitive functions in order to examine the links between cognitive function and attentional control, information processing speed and accuracy.
- The Authors briefly mentioned that the performance in the STAC had been previously compared with the performance in the Flaker task (page 3, line 99). Given that no additional tasks were included to assess the attentional functions of the participants in the present study, I would suggest discussing such findings instead of inviting the reader to look at that paper for further insight. What is the relationship between the attentional constructs investigated by the STAC and those assessed with typical tasks/tests available in literature?
This has been addressed and the following information appears in full: Please see line 118:
Unlike traditional tests, which focus on single elements of attentional control in isolation, STAC is more holistic and integrates components of selective attention, task monitoring, and response inhibition (line 89-97). Furthermore, while there is overlap in the constructs tested, standard tasks such as Flanker (known for its ability to assess response inhibition) require the researcher to set parameters (e.g. stimulus presentation speed) in advance, and often lack flexibility because such values are fixed, meaning the participant is forced to struggle to respond or is not sufficiently challenged. Use of the PEST algorithm as part of STAC, which calibrates speed on the basis of prior responses, has the distinct advantage of ensuring that the task is performed in accordance with an individual’s capabilities. Therefore, the task remains difficult and will challenge the participant as speed is re-adjusted during subsequent PEST cycles (either upwards or downwards, to define their threshold, in the event that the present speed can be exceeded or is too challenging and cannot be sustained).
The comparison of STAC to the Flanker task corresponds to our previous pilot work. While the Flanker task output equates to reaction time (RT), the STAC final speed threshold reflects a participants' ability to perform well under conditions of higher stimulus presentation speeds. Nonetheless, both measures represent efficiency of processing with regard to speed. Where RT from incongruent Flanker trials (representing conditions of maximum difficulty) is assessed in relation to STAC speed, there is strong correspondence between these measures (r(24)=-.650, p=.001). The correlation indicates that as STAC speed increases, Flanker RT decreases, thus signifying performance improvement between the measures is aligned. Therefore, STAC is regarded as comparable to Flanker in relation to outcome measures but has distinct advantages compared to such standard tasks, as outlined above.
- Future studies. The Authors indicated in this section that they intend to implement the task during fMRI scanning. I would suggest the Authors explore the vast electrophysiological (EEG) literature on attentional control. Given the very high temporal resolution of EEG, this technique would be ideal for assessing differences concerning the temporal aspects of the performance. Analysis both in time-domain (using ERPs: P300, CNV component) and in the time-frequency domain would point out the recruitment of additional cognitive resources required for the maintenance of the accuracy with increasing age.
Thank you for this suggestion. It has now been incorporated. Please see line 416:
Additional electroencephalographic [EEG] analysis both in the time-domain and in the time-frequency domain would also reveal the recruitment of additional cognitive resources required for the maintenance of the accuracy with increasing age, together with the examination of the variations in task difficulty an...
Minor points:
- An image that illustrates the timing of the experiment (e.g., one trial) would be more useful instead of a single static image. E.g., a series of frames that illustrate the scrolling up of the left side of the screen, or the target change on the right side of the screen (along with the approximate/mean timing: target change every 19s).
Thank you. This clearer representation (below) is now included. Please see line 224
- A visualization of the results would help the reader in understanding the results. For example, barplots or boxplots could be used for non-parametric analyses, while scatterplots could be used for visualizing the sets of correlation proposed.
Thank you. These (please see below) are now included.
Figure 2. Speed (symbols per minute per column; spm) between young and older adults
Submission Date
12 May 2020
Date of this review
21 May 2020 16:17:34
Figure 3. Correlation between speed (spm) and MOCA score in older adults
Reviewer 2 Report
I was asked to review a manuscript by Torrens-Burton et al. on age-related information processing in a dynamic attentional task. I was very impressed by that study! I think the main aspect that sets this study apart of others in the field is that a new, thought-provoking perspective on age-related information processing is presented that is very likely to advance current views on this matter. The study asks a very intelligent question and uses a very suitable/clever and novel experimental approach to examine attentional control using a dynamic, more ecologically valid measure, the STAC, under the well-derived hypotheses provided by the authors. The logic of the experimental approach is well-explained, making this manuscript also accessible for readers not familiar with this sort of experiments. The line of arguments is well connected to the existing literature. Generally, the methods are sound, the behavioral analyses adhere to the highest standards and I could not identify any important weakness in the data analyses. The results seem reliable, and are easy to follow, and possible limitations are discussed. The discussion is well balanced. I think that this is a strong paper for publication in Brain Sciences and I think that the manuscript fits well into the profile of the journal. However, I have a minor point the authors should address in a revision.
- A significant correlation was found between spm and MoCA score [r = .25, p = .018]. Would this still be significant after correction for family-wise error, given the large number of dependent measures employed in the study? And if not, how would this alter the conclusions of the study?
Author Response
Thank you very much for reviewing our paper and for your very kind comments.
Regarding the Bonferroni correction. Thank you for raising this issue and highlighting things we should have made clearer in our manuscript. This is very much an issue we have considered, which led us to taking advice from a statistician, with respect to whether or not it should be used in our study. We were advised not to use this correction for various reasons, namely; ‘such a correction should be used primarily when the researcher is looking for significant associations but without a pre-specified hypothesis’. In line with this we had already set out a-priori hypotheses, namely the first family (subset) or major area of interest (with an a-priori hypothesis) was between spm and MoCA and spm and age. This was designed to give us our predicted dichotomy. The other family or subset consisted of more exploratory factors (MFQ, NART, anxiety/depression) and as there was no significant correlation between spm and any of these factors, Bonferroni correction was not applied because of the risk of a string probability of a type II error occurring. Within the manuscript we have now tried to make the main aims of the study clearer.
Round 2
Reviewer 1 Report
I truly appreciated the effort of the Authors in considering all the points raised during the previous round of revision. The changes in all sections of the manuscript and the addition of figures increased the work's readability and fruition.